# LiDAR and UAV SfM-MVS of Merapi Volcanic Dome and Crater Rim Change from 2012 to 2014

Christopher Gomez [1,2,*], Muhammad Anggri Setiawan [2,3], Noviyanti Listyaningrum [2,3], Sandy Budi Wibowo [4], Danang Sri Hadmoko [2,3], Wiwit Suryanto [5], Herlan Darmawan [5], Balazs Bradak [1], Rikuto Daikai [1], Sunardi Sunardi [6,7], Yudo Prasetyo [8], Annisa Joviani Astari [9], Lukman Lukman [10], Idea Wening Nurani [11,12], Moh. Dede [7], Indranova Suhendro [2,3], Franck Lavigne [12] and Mukhamad Ngainul Malawani [3,12]

1 Laboratory of Sediment Hazards and Disaster Risks, Faculty of Oceanology, Kobe University, Kobe 657-8501, Japan
2 Center for Disaster Study, Universitas Gadjah Mada, Yogyakarta 55284, Indonesia
3 Department of Environmental Geography, Faculty of Geography, Universitas Gadjah Mada, Yogyakarta 55281, Indonesia
4 Department of Geographic Information Science, Faculty of Geography, Universitas Gadjah Mada, Yogyakarta 55281, Indonesia
5 Faculty of Mathematics and Natural Sciences, Universitas Gadjah Mada, Yogyakarta 55281, Indonesia
6 Faculty of Mathematics and Natural Sciences, Universitas Padjadjaran, Sumedang 40133, Indonesia
7 Doctoral Program on Environmental Science, Postgraduate School (SPs), Universitas Padjadjaran, Bandung 40132, Indonesia
8 Department of Geodetic Engineering, Faculty of Engineering, Universitas Diponegoro, Semarang 50277, Indonesia
9 Geographic Information Science Study Program, Faculty of Social Science Education, Universitas Pendidikan Indonesia, Bandung 40154, Indonesia
10 National Research and Innovation Agency (BRIN), Cibinong 16911, Indonesia
11 Department of Development Geography, Faculty of Geography, Universitas Gadjah Mada, Yogyakarta 55281, Indonesia
12 Laboratory of Physical Geography UMR 8591, Université Paris 1 Panthéon-Sorbonne, 94320 Thiais, France
* Correspondence: christophergomez@bear.kobe-u.ac.jp

**Abstract:** Spatial approaches, based on the deformation measurement of volcanic domes and crater rims, is key in evaluating the activity of a volcano, such as Merapi Volcano, where associated disaster risk regularly takes lives. Within this framework, this study aims to detect localized topographic change in the summit area that has occurred concomitantly with the dome growth and explosion reported. The methodology was focused on two sets of data, one LiDAR-based dataset from 2012 and one UAV dataset from 2014. The results show that during the period 2012–2014, the crater walls were 100–120 m above the crater floor at its maximum (from the north to the east–southeast sector), while the west and north sectors present a topographic range of 40–80 m. During the period 2012–2014, the evolution of the crater rim around the dome was generally stable (no large collapse). The opening of a new vent on the surface of the dome has displaced an equivalent volume of $2.04 \times 10^4$ m$^3$, corresponding to a maximum $-9$ m ($+/-0.9$ m) vertically. The exploded material has partly fallen within the crater, increasing the accumulated loose material while leaving "hollows" where the vents are located, although the potential presence of debris inside these vents made it difficult to determine the exact size of these openings. Despite a measure of the error from the two DEMs, adding a previously published dataset shows further discrepancies, suggesting that there is also a technical need to develop point-cloud technologies for active volcanic craters.

**Keywords:** Merapi Volcano; Indonesia; natural hazards; disaster risk; point-cloud technology

## 1. Introduction

On stratovolcanoes, domes are a major source of hazards as they often collapse under the actions of both internal gas pressure [1] and gravity [2], creating hazardous pyroclastic-density currents [3]. Without gravitational collapses, internal gas pressure can generate explosive eruptions, eventually propelling ash and other volcanic material into the upper atmosphere. In addition, chemically stable domes can still explode as a result of phreatic and phreatomagmatic processes [4]. Even during the more quiescent phase, volcanic domes are still evolving, eventually sliding away from the top of the volcano [5] and breaking apart into "smaller pieces", generating long-runout rockfalls [6]. Environmental factors of the dome also contribute to the dome instability: for instance, precipitation contributing to hydrothermal alteration; the general movement of the volcanic structure (uplift and subsidence); the local and regional seismic activity (from Vazquez et al. after the work of McGuire [7]). However, as it was recently pointed out, despite a variety of explaining factors, the contribution of one over another remains scientifically unclear [8]. For disaster risk management and to avoid catastrophe, dome monitoring is thus essential, even during more quiescent periods.

In Central Java, Indonesia, a series of dome-collapse pyroclastic-density currents have been sweeping the flanks of Merapi Volcano during the Holocene period, although larger eruptions and sector collapses were also found in the earlier Quaternary period, known since ~360,000 BP [9,10]. The pyroclastic-density currents of the last major eruption in 2010 (VEI4: [11]) covered an area of 22.3 km$^2$ [12,13], in turn turning into lahars that have flooded the valleys towards Yogyakarta City [14,15]; in its aftermath, the growth of the dome has also generated rockfall hazards [6].

Consequently, the evolution of the dome has attracted the attention of scientists in different fields of research, e.g., rock geochemistry [16], gas analysis [17], numerical modeling [18], petrology [19], surface deformation [20,21], and seismology [22]. During the historical period, the dome of Merapi Volcano has been lodged into a horseshoe crater-rim, opened towards the south, directing most of the gravity collapse pyroclastic flows and rockfalls [6], locally called "*guguran*", in this same direction (Figure 1).

The present dome was born from the millennial eruption of 2010 [13], which topographically resembles a 150 m-diameter tabletop, which fractured due to phreatic explosions between 2012 and 2014 [4]. Monitoring of volcanoes and dome evolution represents significant technical challenges and risks for the personnel. Therefore, scientists have been striving to create models and simulations of dome growth and collapse [8], but field data and evidence are still essential to support those models. Focusing on the period 2012–2014, the present contribution proposes the use of high-resolution geodetic measurements from airborne LiDAR and UAV photographs for photogrammetric purposes, in order to detect topographic changes that would have occurred in the crater rim during this period of relative quiescence.

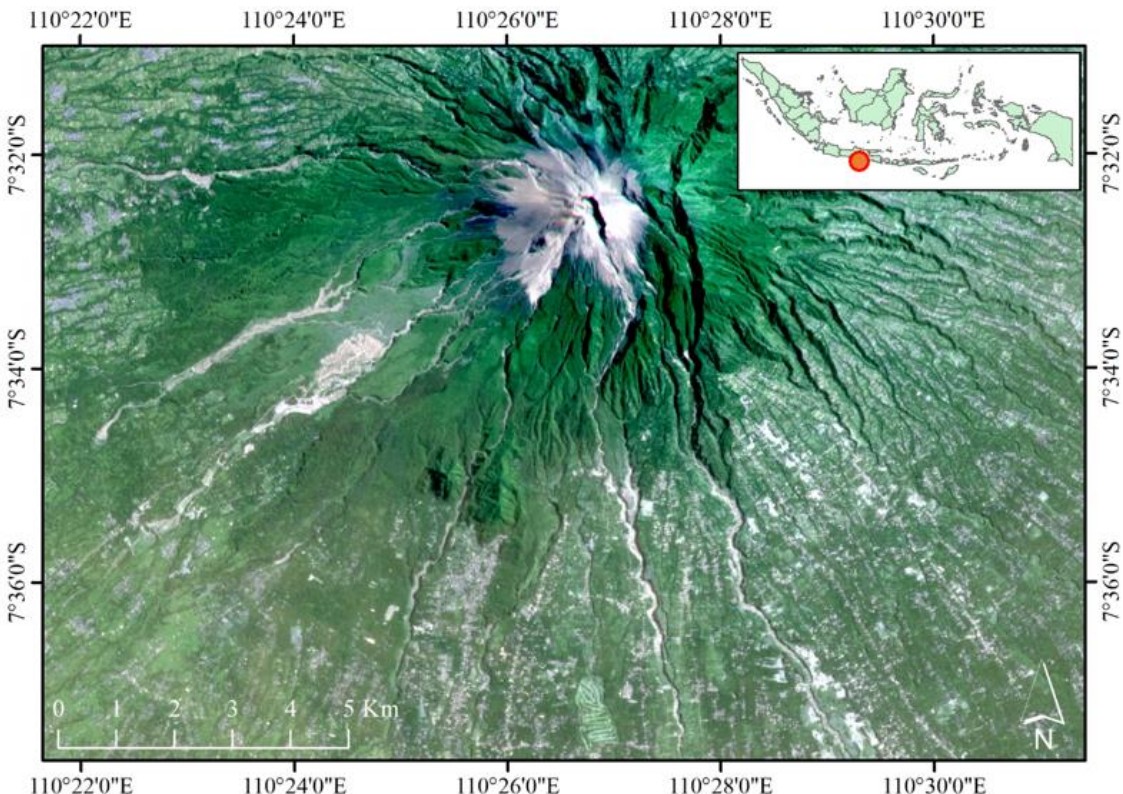

**Figure 1.** High-resolution (3 m) image from the PlanetScope satellite for the area of Mount Merapi and its surroundings displayed with a true color composite. Dwellings and agricultural land as close as 5 km from the dome translate into high-level disaster risk.

## 2. Materials and Methods

The present contribution includes two sets of data: one LiDAR-based dataset from 2012 and one UAV dataset from 2014. The 2012 dataset was derived from airborne LiDAR with a density of >5 point/m$^2$. The LiteMapper 5600 System was installed on a Cessna 402B Aircraft flying at an altitude of 820 m above the summit, and the photograph's airspeed was 259 km/h. The side overlap and the frontal overlap were, respectively, 40% and 60%. Concomitantly, GPS surveying for LiDAR base station was conducted at Badan Informasi Geospatial (BIG) from reference points using RTK GPS Trimble R9 with a minimum of 6 satellites. From this dataset, the 2012 data were gridded at a 1 m horizontal resolution, using the minimum vertical value (elevation) in each square-meter grid.

The 2014 dataset was built using structure from motion from 328 photographs of 3000 × 4000 pixels. The photographs were acquired on 16 October 2014 between 12:30 and 12:50 using a fix-wing UAV mounted with a Canon PowerShot S100 at a focal length of 5 mm, an exposure time of 1/1250 s, and an ISO speed of ISO-80. The photographs were integrated into the SfM–MVS (structure-from-motion multiple-view stereophotogrammetry) software Metashape Pro, commercialized by Agisoft. The process includes point-cloud reconstruction and its densification. The dense point cloud was then exported to Cloud Compare, and the point cloud was subsampled at 1 point per square meter to exactly match the point location of the LiDAR data. Using this process, the next step of comparing the two datasets limits the importance of the artifacts linked to variable point density and the horizontal distance between points. The construction and the modalities of the combination of these two datasets are as follows:

The two point clouds are then aligned in Cloud Compare (open-source software) to scale the SfM-MVS data and match it to the LiDAR point cloud using the C2C (Cloud to Cloud) algorithm, from which the distances in the x, y, and z directions are separated.

The LiDAR was thus used as the "true elevation" to calibrate the UAV photogrammetry, although errors varying based on the type of surface have been reported to vary between 18.9 cm for pavement and 25.9 cm for deciduous tree elevation [23]. Moreover, LiDAR error increases with the slope, and at Mt. Erebus, for instance, for a slope of 20 degrees, where an additional vertical error of 16 cm had to be added, locally reaching 21 cm [24]. In the present case, the authors chose to double those error values and estimate that the LiDAR was accurate to about 50 cm and that any change below this value may not be representative (although all values are reported in the article). The choice to double this value was also motivated by the possibility of sand and ash grains bouncing near the surface, increasing the potential for error.

It resulted in an RMSE (root mean square error) of 90 cm between the two point clouds when not taking into account the crater area that was known to have changed. Adding both the error from the aligned 2014 point cloud to the error assigned to the LiDAR data, the elevation error for the measurements comparing the two surfaces is about 1.4 m.

The dataset from which the authors have worked is thus made up of an orthophotograph of the summit in 2014 (Figure 2a), and two datasets of the dome in 2012 (Figure 2b) and 2014 (Figure 2c). From a visual inspection of the hill-shaded representation of the DEM, one can see the changes that have occurred on the dome (most notably the opening of vents [4]).

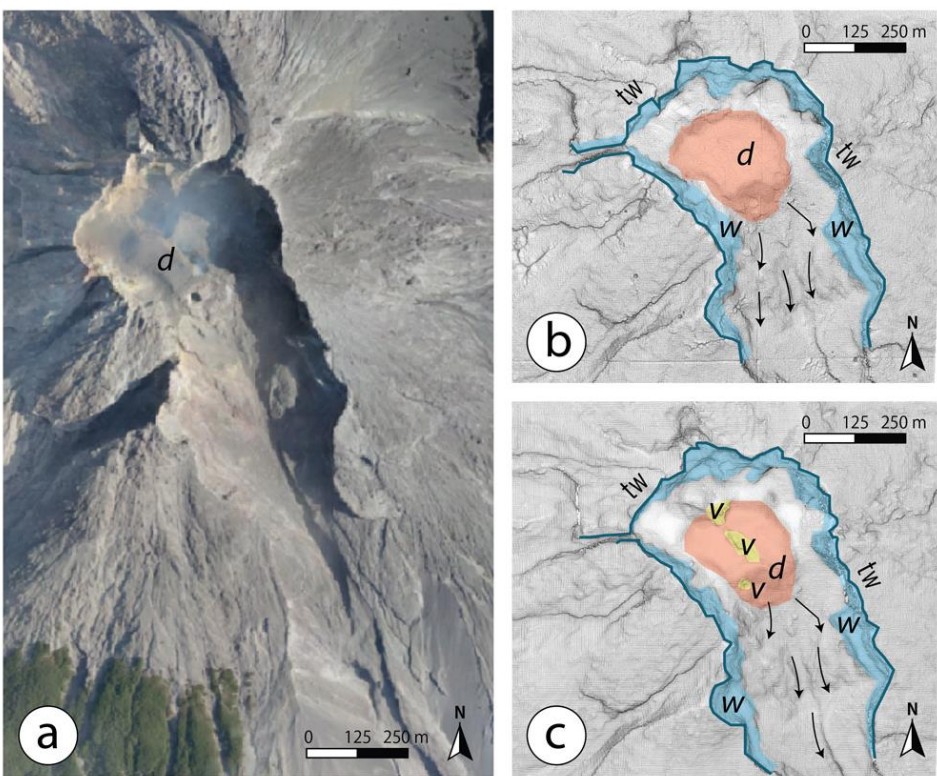

**Figure 2.** Dome of Merapi Volcano in 2012 and 2014: (**a**) orthophotograph constructed from UAV imagery in October 2014; (**b**) surface map of the dome in 2012 from LiDAR and (**c**) in 2014 from UAV-SfM-MVS. The captions are (d) dome in transparent red; (v) vent in transparent yellow; (tw) top of the wall and it corresponds to the blue line; (w) wall and it corresponds to the transparent blue.

## 3. Results

The comparison of the two topographic data is presented with first the modifications to the crater rim and the talus, and secondly, a quantification of the volume removed either from the volcanic explosions or gravitational collapses.

### 3.1. The Crater Rim and Its Talus

At the summit of Merapi Volcano, a horseshoe crater rim traps the dome. During the period 2012–2014, the crater walls were 100–120 m high above the crater floor at its maximum (from the north to the east–southeast sector), while the west and north sectors present a topographic range of 40–80 m (Figure 3). The crater is open to the south. During the period 2012–2014, the evolution of the crater rim around the dome was generally stable (no large collapse), although the data shows variation in the geometry of the subvertical walls around the crater rim (b–h, and k in Figure 3). The walls that spread outside of the crater rim seem to be more stable (a,l, and m in Figure 3). At the foot of the crater walls, the topography from 2012 to 2014 was stable overall, showing no major wall collapses, except in e, h, and I (Figure 3), where the floor level increased by >2 m locally. Although the walls seem to show locally large discrepancies between 2012 and 2014 (Figure 3b), the absence of significant deposits at their toes suggests that those are potential artifacts that are examined in the discussion.

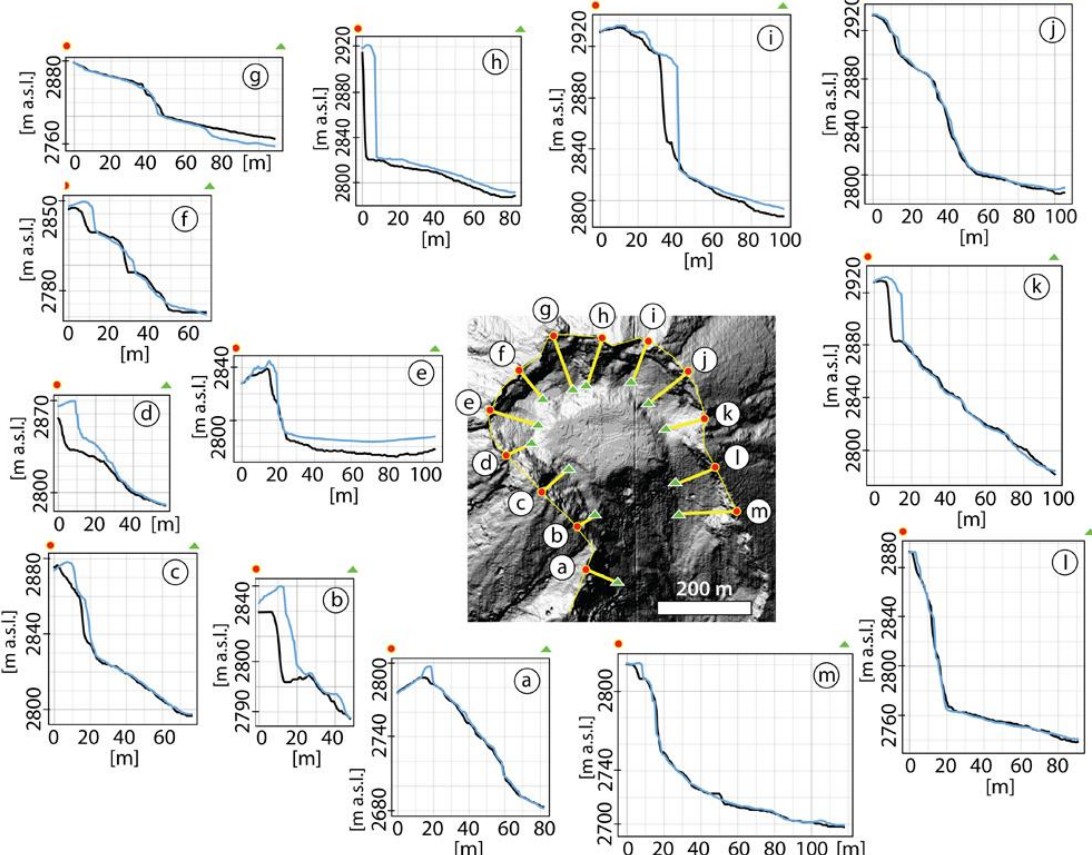

**Figure 3.** Cross-sections of the crater rim. The black line is the topography in 2012 as extracted from the LiDAR-DEM at 1 m horizontal resolution, and the blue line is the topography in 2014, extracted from the UAV-based photogrammetric model.

### 3.2. Dome's Topographic and Volumic Change with the Localized Explosions

As already reported in the literature, during the period 2012–2014, the dome locally exploded and split into a set of aligned new vents along a north–west–southeast axis. Around the dome, surface changes also occurred with an increase in elevation (Figure 4). The opening of a new vent on the surface of the dome has displaced an equivalent volume of $2.04 \times 10^4$ m³, corresponding to a maximum $-9$ m (+/−0.9 m) vertically (location 4 in Figure 4). At the two vents generated to the north and south of the dome, the volume changes are $8.56 \times 10^4$ m³ at location 1, $2.61 \times 10^5$ m³ (location 2), and $4.82 \times 10^3$ m³ at location 5 (Figure 4). The variability is mostly controlled by the visual depth of the

openings (if obstructed by debris, they appear shallower). Therefore, those values need to be considered as minimal values. Surrounding the plateau created by the dome, loose material is also displaying topographic change. It has mostly increased this time. For the period 2012–2014, the rise of material corresponds to a respective volume change of $1.88 \times 10^4$ m³ at location 3 and $6.22 \times 10^4$ m³ at location 8. These changes correspond to changes nearing 18 m $(+/- 0.9$ m) at location 1 and 11 $(+/- 0.9$ m) m at location 8. On top of these major topographic changes, gravitational collapses have also been observed on the crater rim (location 10) and a small portion to the south of the dome seems to have also collapsed (location 6 in Figure 4).

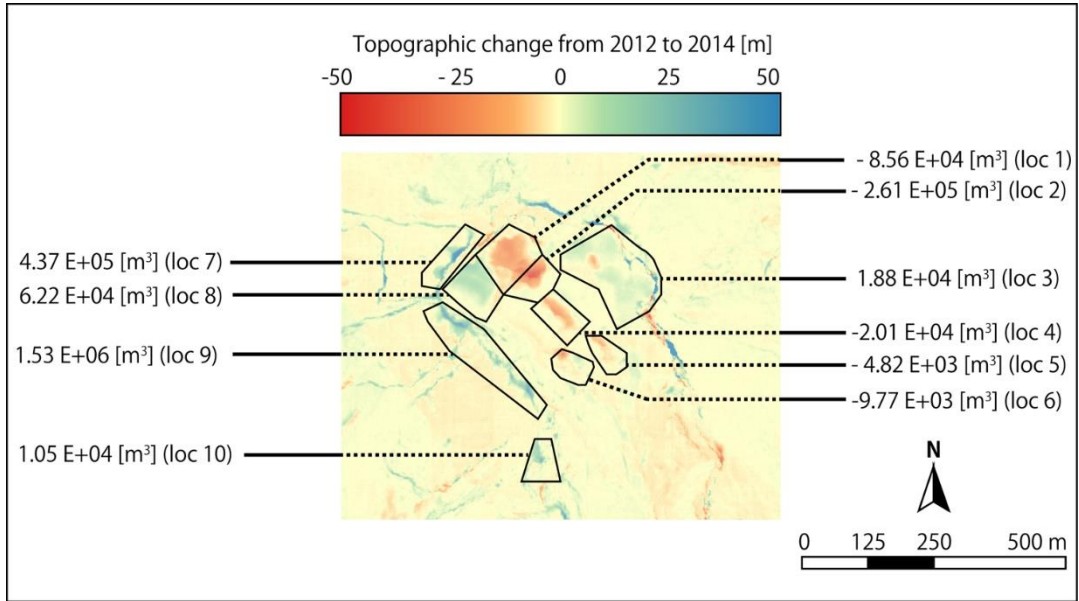

**Figure 4.** Topographic change from 2012 to 2014. These data show the vents opening and the "split in two" of the dome, as well as the areas between the crater rim and the dome, where the topography increased, arguably due to exploded material deposition. These changes emphasize the changes that are >2.5 m (i.e., in blue on the map), valued above the overall RMSE and the RMSE of the outside structure. This procedure may erase some of the minor variations, but it is aimed at eliminating the false positives from the dataset.

## 4. Discussion

This discussion is organized into two sections, with the first being an interpretation of the results and what they mean in terms of volcanic processes. Then, it follows a section that compares the data from the present contribution with a dataset from 2015, and finally we discuss the implications of the present work for hazards and disaster risk monitoring.

### 4.1. Surface Deformation and Interpretation

The dome of Merapi Volcano locally exploded during the period 2012–2014, creating a set of aligned vents [4], which represent an estimated volume of material of $2.01 \times 10^4$ m³ for the central elongated vent at the top of the dome, $2.61 \times 10^5$ m³ to the north (high-value determined by good visibility at a depth of almost 30 m below the pre-explosive surface), and one vent to the south, just shy of $1 \times 10^4$ m³. The central elongated vent represents a topographic drop of about 9 m (Figure 5), although it is most likely that the opening could have been deeper (filled by either ejecta going back to the vent or the full depth being invisible to the SfM-MVS method due to the narrowness of the vent). The origin of these explosions is the result of pressure increase from the contact with rainwater [4] as well as probable material weakening due to hydrothermal alteration, a process that was observed at the dome summit and its surroundings at a later date [25]. Around the dome, the material in at least two locations rose topographically (Figure 5), even without having had major

wall collapses that would match this rise. If we work by analogy, the dome of 2011, which is visible between 2012 and 2014, may have grown over other extruded material in the same way that the 2017 dome climbed over the 2011 dome [25]. The exploded material may have partly escaped the space of the crater rim, but it is most likely that a large portion of it remained inside the crater, with the largest blocks traveling a shorter distance. Despite a measure of the visible volume change (Figure 4), it is difficult to link both the post-phreatic explosion holes to the deposits for the following reasons: (1) Due to decompression and deposit bulk density changing from that of the dome, the measured volume change does not reflect a material volume change; (2) Part of the material escaped the crater rim; (3) The post-explosion holes will have certainly collapsed; (3) For the deepest part of the holes, ALS or SfM-MVS may not record the real "bottom". This shows the necessity to develop a methodological framework that is more adapted to volcanic vent surveying due to the numerous complexities it holds.

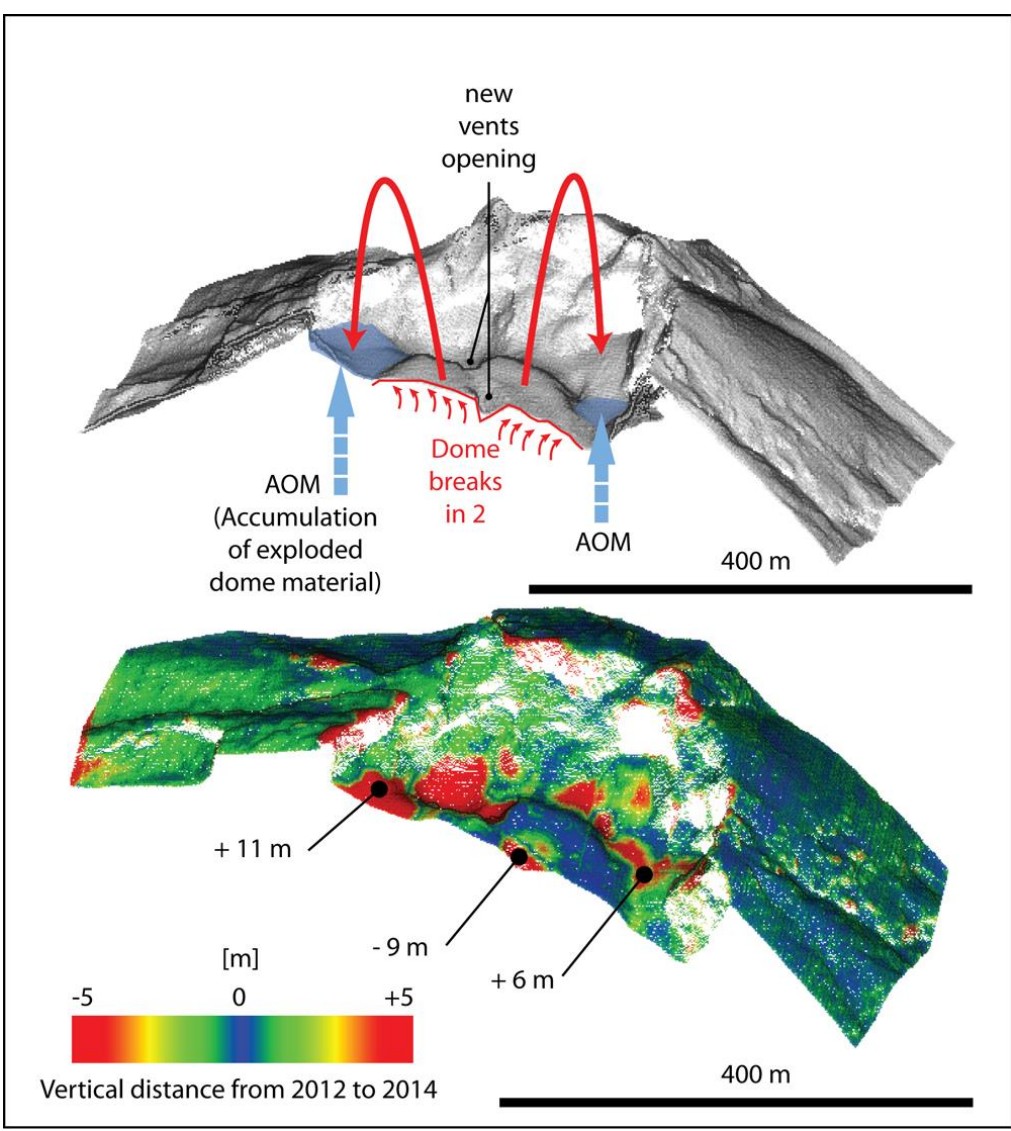

**Figure 5.** Interpretation figure of the deformation at the dome and near the dome.

### 4.2. Hazards in Quiescent Time, When the Volcano Acts as a Mountain

Even in periods when the volcano is not magmatically active [25], minor explosions and other erosion processes can also occur. In the present case, there is an accumulation of volcanic clasts within the crater and surrounding the volcanic dome. Such low-energy events can still be a hazard to mountain climbers, as has been seen at Mt. Ontake [26],

where, in 2014, around 340 individuals were located in the surrounding of the summit when it exploded, creating ballistic for which hazard maps are essential [27]. Based on the topographic change at Merapi Volcano, a large portion of the large ballistic seems to have been contained within the crater, as the topographic change from 2012 to 2014 does not show significant accumulation zones outside the crater. However, even if the material is contained within the crater and the crater is defined as a "no trespassing zone", the material produced usually ranges from fine-grain material to coarser material, and upon rainfall, the material can also be transformed into a slurry of debris and water, eventually creating lahars, such as when it did for the Ontake eruption [28]. These "small-scale" hazards can then be very pervasive because, instead of imposing a "new rhythm" on the volcano and the valley, they will be highly dependent on other factors, such as a trigger by an earthquake, or in some cases, the role of seasonal snow cladding [29]. Finally, unconsolidated material located at the summit of the volcano is a further source of hazard during long quiescent times, as it has been shown at Unzen Volcano [30], where the gullies are progressing towards the volcanic dome by regressive erosion, modifying the watershed geometries [31] and helping the release of material from the summit. In the case of a solid compact dome, breaking parts may require some "extra-effort", but if the material is already unconsolidated, it is a further pool of material that can be released on the slopes.

From a hazard perspective, small-scale explosions of a phreatic type only affect a small area, but the timing of such an explosion is extremely difficult to predict, and the fragmentation of material results in a pool of clastic sediments at the top that can then be remobilized by seasonal events (rain and snow melting) as well as longer-timescale events such as regressive erosion or eventual seismic acceleration.

**5. Conclusions**

The main conclusions of the present paper are that (1) during the period 2012–2014 when the dome of Merapi experienced phreatic explosions, the topography around the dome rose; (2) this rise does not seem to be related to large wall collapses, and it is likely that the exploded material accumulated in a low-topographic area; and (3) from a technical perspective, there is a need to develop UAV-imagery-acquisition methods that are fit for volcanic craters, with step walls (e.g., avoiding NADIR images), in order to reduce the error due to the lack of data; (4) volcanoes, even during quiescent time, are hazardous and for this reason, inter-eruption monitoring is essential.

**Author Contributions:** Conceptualization, C.G., F.L., S.B.W., D.S.H., M.N.M.; methodology, M.A.S., N.L., B.B., R.D.; resources, M.A.S., W.S., S.S., Y.P., A.J.A., L.L., I.W.N., M.D.; writing—original draft preparation, C.G.; writ-ing—review and editing, C.G., F.L., H.D., S.B.W., I.S.; funding acquisition, S.B.W., C.G. All authors have read and agreed to the published version of the manuscript.

**Funding:** This research was funded by Riset Kolaborasi Indonesia (RKI) number 1545/UN1/DITLIT/Dit-Lit/PT.01.03/2022. The authors also thank two anonymous reviewers and the editor for their valuable comments to strengthen this paper. Further funding was provided at Kobe University through the IMARC international program on volcanic hazards, led by C. Gomez.

**Data Availability Statement:** All the data used in the present contribution can be accessed upon contacting the corresponding author.

**Conflicts of Interest:** The authors declare no conflict of interest.

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
