# Peer review of "LiDAR and UAV SfM-MVS of Merapi Volcanic Dome and Crater Rim Change from 2012 to 2014"

_remotesensing, doi:10.3390/rs14205193_

Round 1

Author Response

Dear Reviewer 1:

First of all thank you for your review, you spotted the important issue in this paper, and I have to agree with all the points you made, on the clarity of the objectives, the issue of using inappropriate vocabulary like "deformation" instead of topographic change, as well as the difficulty on reading some of the figures.

With the answers to your comments (underneath), I am also sending you a version of the new manuscript with the changes "in red", and for you to also see the changes in the figure (notably the one that was based on the DoD method, which does not show the direction of change of the topography).

Finally, for the last figure (figure 6) that was using an external dataset, and which we were using to show that regardless of the improvement in the topographic change measurement methods, there are still a lot of work to be done, I was not able to make changes to your satisfaction I think, and in hindsight, it is tackling another issue than the one proposed in the present manuscript. We therefore decided to keep this "new issue" out of the paper this time and we will work on it and make it a proper research topic. Instead of this topic, which was maybe not the best choice for the discussion, we have changed it to the "hazard implication" you suggested, as during more quiescent time, "other types" of hazards may take the precedence (like we have seen at Ontake in Japan for instance), and we thus discussed the importance of the results from this perspective.

Once again, thank you for your comments and suggestions.

You will find the answer to your comments underneath, and we are also attaching a version of the manuscript with the new text in red.

The authors have collected an impressive data set for Merapi, but the analysis of these data lacks focus and the presentation of results lacks clarity. This manuscript will require major revision before I can recommend that it be published.

What was the objective of this research? Documenting changes in dome and crater morphology over time? Linking these changes to magmatic and/or phreatic and mechanical (non-magmatic) processes? Differentiating between magmatic and non-magmatic processes based on changes on morphology? Readers may infer objectives such as these, but the authors need to express the objectives explicitly.

(ANSWER) I agree with this assessment of the clarity of the objectives, even the abstract was announcing some objectives and then presenting results from a rather different light. The objectives have been now announced as the documentation/measure of topographic change, and all the part of the magmatic processes has been removed, including the usage of “deformation”, because it is not sustained by any of the data we are presenting here.

This problem appeared in the main body of the text and in the abstract.

The abstract has been changed (red in the document attached), because it was (unfortunately) a condensed of the issues that the reviewer has identified (lack of clear objective) and lack of clarity in the exposition of the data. It is now a better representation of the content of the manuscript. The write-up was actually showing a discussion we were having on whether the dome had been growing from inside or was created by the fallout from the explosion, or a combination of both. And, because there had been no major seismic activity that would be associated with a magma injection, we decided to go with the fallout from the explosion (which makes much more sense because the dome did not rise and only the surrounding that are topographically “low” = accumulation zone). Based on the reviewer’s suggestion, we also used more “neutral” vocabulary such as topographic change instead of wording suggesting an internal growth, which we actually can’t sustain with any data.

Adding to this lack of focus, the authors appear to consider any change in topography as deformation of the dome and crater. It is more useful to restrict the use of “deformation” to the growth of the dome by endogeneous and exogenous (i.e., magmatic) processes. In my opinion, the most compelling aspect of this research is the potential to differentiate magmatic and non-magmatic processes based on morphologic changes over time. The analytical results, as presented, suggest that non-magmatic (phreatic and mechanical) processes were dominant during the time period of the study (2012-2015). This interpretation would be useful in an assessment of volcanic hazards.

(ANSWER) I also agree with this concern and the inadequate vocabulary was removed. The point made on quiescent-time morphological change is indeed important and interesting. I have added a point (yet short) in the discussion. Thank you.

Figures 4 and 5 lack clarity, as the authors have chosen to represent both positive and negative changes in topography with the same color schemes. I realize that the goal was to highlight changes, regardless of the direction of the change, but this presentation does not discriminate between the negative changes due to the opening of vents and positive changes due to the accumulation of debris due to phreatic (and gas) explosions and mechanical failure of the dome and crater walls. The authors do label the direction of change at a limited number of locations, but a different color scheme that identifies positive and negative changes explicitly would be very helpful.

(ANSWER) Using the DEM of differences as absolute differences was certainly not the best choice in the present case, as the topography changes in both direction over time. We have thus remade the figure, by modifying the color-scale, in order to reflect the direction of the change. The “neutral” blue was also replaced by a pale color, so that the figure does look less “overwhelming” and is easier to read. Thank you.

The authors introduce a third data set, collected in 2015, in the Discussion Section (Section 4), with no citation of the source of these data. The caption for Figure 6 refers to Damarawan et al., (2018), but this citation belongs in the main body of the text. In addition, the comparison of the 2015, 2014, and 2012 data (Fig. 6) is confusing. Rather than compare the 2015 data along the same transects shown in Figure 3, the multiple data sets are compared along a set of transect radiating from the “center” of the dome (Fig. 6). The authors present no other information on the locations of the radial transects, preventing any direct comparison between Figures 3 and 6. The lack of direct comparisons is critical, as Figure 6 indicates virtually no changes between 2012 and 2014 in the majority of the transects. However, neither Figures 3 nor 4 use consistent vertical scale on the plots. The transects must be plotted on the same vertical scale, both within and between the figures, to allow any meaningful comparison between the 2012, 2014, and 2015 survey results. In conclusion, I repeat that the manuscript requires major revision before consideration for publication. If the authors are not prepared to do such revision, then this manuscript should be rejected

(ANSWER) This figure 6 and portion of the discussion has been removed now. On figure 6, I should have added a small map showing where the transects are, this would have been much better, but as for each transect I provide 2012, 2014 and 2015 I did not think that it would be a problem. Now, I tried to find data on the error analysis and the protocol for the 2015 data but these looks rather difficult to access, and in the absence of those, I think that figure 6 is just adding confusion rather than anything else.

The original idea was to say that even with LiDAR data and photogrammetry data with a measure of the error and a numerical value on the accuracy, adding further dataset can be a challenge on its own.

In hindsight, it does not bring anything to the paper. Therefore, in the discussion, instead of this section, I have written about the point that the reviewer made earlier on “quiescent time hazard implication of topographic change”.

Reviewer 2 Report

Dear editor and authors,

I have read the manuscript "LiDAR and UAV SfM-MVS of Merapi Volcanic Dome And Crater Rim Change from 2012 to 2014" with great interest. The authors present a comparative study of derived topographic data of Merapi volcano, obtained in 2012 and 2014 through two different methods (one LiDAR-based dataset and one obtained via UAV-based photogrammetry). The study neatly demonstrates the evolution of the summit of Gunung Merapi in the intervening years, supplemented by the inclusion of data from an earlier study (Darmawan et al. 2018). This comparative analysis illuminated areas of relative uplift and subsidence, which the authors attribute to shallow eruptive phenomena, including the build-up of exploded dome material. The authors also determine an area that will likely be subject to collapse in the future, which is a key objective of this kind of remote sensing work. All in all, I heartily recommend publication of this work, which is a good demonstration of the value of this method. I have a few (extremely minor) comments and questions that might help improve the paper, listed here:

1. Line 34: "aime" should be "aims"
2. Line 183: Please remove "This is a figure." There is also an extraneous full stop in this caption, between "from" and "the".
3. Line 196: "shallows" should read "shallower" or "to be more shallow"
4. Figure 4: For accessibility, the authors could consider using a perceptually uniform colourmap here.
5. Line 214: "the authors are discussing" could be "we discuss"
6. Line 226: "processed" should be "process"
7. Line 236: Suggest replacing "the one of" with "that of" for clarity.
8. Line 237: No need for the "and" here.
9. Line 238: "collapse" should be "collapsed" for the correct tense.
10. Line 239: No need for "of those"; suggest removing and replacing "it" (line 238) with "the holes" or similar.
11. Line 241: Suggest removing "that's why".
12. Figure 5: The colourmap here precludes the reader from knowing whether displacement is positive or negative. Suggest replotting with a diverging colourmap (e.g. "vik" or "roma"
https://www.fabiocrameri.ch/colourmaps/).
13. Line 265: Typo in Darmawan.
14. Figure 6: The orange and green arrows are quite distracting--I would recommend removing these entirely as the data speaks for itself. The caption states that the arrows indicate "significant difference" in topography, but it isn't apparent that this is based on any statistical (or otherwise quantitative) assessment. There are also double-headed arrows which are not explained. A good addition to the figure could be a small clock face/compass symbol in each panel illustrating the azimuth. More generally, it seems that most of the supposedly "significant" displacements are not discussed by the authors. There is an exception: "[A]
t 80 degrees [...] the topographic wall is moving inward with large blocks with a crack in its center appearing. This large portion of the subvertical wall is likely to collapse onto the crater floor and the dome". It would be terrific if the authors could supplement this description with photographic evidence. Similarly, some mention (and more detailed drone imagery) of other regions in the study area showing marked differences (e.g. as highlighted in Figure 5) would be a valuable addition to the paper, I think. As the authors have hundreds of drone images, it seems a shame not to show some here.
15. Line 288: "advices" should be "advice".
16. Line 325: Typo ("Volcnaol")

Author Response

To the reviewer.

Thank you for your constructive comments and for the guidance you also provided to solve the issues you have raised, including the English typos and errors.

The authors have now made the necessary changes, however, you will notice "other" changes based on reviewer 2 comments. So that, two of the figures have also changed based, and the section comparing the two datasets to a third one that was previously published appeared to bring more problem than any improvement to the manuscript, it was thus deleted. The data construction comparison may needs more dedicated dataset to be solved adequately.

  1. Line 34: "aime" should be "aims"

Done
2. Line 183: Please remove "This is a figure." There is also an extraneous full stop in this caption, between "from" and "the".

Done
3. Line 196: "shallows" should read "shallower" or "to be more shallow"
4. Figure 4: For accessibility, the authors could consider using a perceptually uniform colourmap here.
5. Line 214: "the authors are discussing" could be "we discuss"

Done
6. Line 226: "processed" should be "process"

Done
7. Line 236: Suggest replacing "the one of" with "that of" for clarity.

Done
8. Line 237: No need for the "and" here.

Done (removed)
9. Line 238: "collapse" should be "collapsed" for the correct tense.

Done
10. Line 239: No need for "of those"; suggest removing and replacing "it" (line 238) with "the holes" or similar.

Done both removal and replacing it by “the holes”.

  1. Line 241: Suggest removing "that's why".

This line was removed as the next section has been removed and changed.

  1. Figure 5: The colourmap here precludes the reader from knowing whether displacement is positive or negative. Suggest replotting with a diverging colourmap (e.g. "vik" or "roma" https://www.fabiocrameri.ch/colourmaps/).

This information has been added to figure 4, which originally was only the DoD. It is now possible to see whether the ground level has increased or decreased. Therefore, the authors made the choice to keep this figure 5 as it was. Would you have a rebuttal against this decision, changes can still be made.

The end of the manuscript was redesigned, as per suggestion of reviewer 1.

  1. Line 265: Typo in Darmawan.
    14. Figure 6: The orange and green arrows are quite distracting--I would recommend removing these entirely as the data speaks for itself. The caption states that the arrows indicate "significant difference" in topography, but it isn't apparent that this is based on any statistical (or otherwise quantitative) assessment. There are also double-headed arrows which are not explained. A good addition to the figure could be a small clock face/compass symbol in each panel illustrating the azimuth. More generally, it seems that most of the supposedly "significant" displacements are not discussed by the authors. There is an exception: "[A]t 80 degrees [...] the topographic wall is moving inward with large blocks with a crack in its center appearing. This large portion of the subvertical wall is likely to collapse onto the crater floor and the dome". It would be terrific if the authors could supplement this description with photographic evidence. Similarly, some mention (and more detailed drone imagery) of other regions in the study area showing marked differences (e.g. as highlighted in Figure 5) would be a valuable addition to the paper, I think. As the authors have hundreds of drone images, it seems a shame not to show some here.
    15. Line 288: "advices" should be "advice".

Sentence removed
16. Line 325: Typo ("Volcnaol")

Done

Once again, thank you for your review. I am sending a copy of the pdf with the changes in red.

Round 2

Reviewer 1 Report

Thank you for your attention to my comments and recommendations.